# *Piper nigrum* CYP719A37 Catalyzes the Decisive Methylenedioxy Bridge Formation in Piperine Biosynthesis

**DOI:** 10.3390/plants10010128

**Published:** 2021-01-09

**Authors:** Arianne Schnabel, Fernando Cotinguiba, Benedikt Athmer, Thomas Vogt

**Affiliations:** 1Leibniz Institute of Plant Biochemistry, Department Cell and Metabolic Biology, Weinberg 3, D-06120 Halle (Saale), Germany; arianne.schnabel@ipb-halle.de (A.S.); benedikt.athmer@ipb-halle.de (B.A.); 2Instituto de Pesquisas de Produtos Naturais (IPPN), Universidade Federal do Rio de Janeiro (UFRJ), Avenida Carlos Chagas Filho, 373, 21941-902 Rio de Janeiro/RJ, Brazil; Fernando@ippn.ufrj.br

**Keywords:** black pepper, cytochrome P450, enzyme activity, methylenedioxy bridge, piperine, *Piper nigrum*, yeast expression

## Abstract

Black pepper (*Piper nigrum*) is among the world’s most popular spices. Its pungent principle, piperine, has already been identified 200 years ago, yet the biosynthesis of piperine in black pepper remains largely enigmatic. In this report we analyzed the characteristic methylenedioxy bridge formation of the aromatic part of piperine by a combination of RNA-sequencing, functional expression in yeast, and LC-MS based analysis of substrate and product profiles. We identified a single cytochrome P450 transcript, specifically expressed in black pepper immature fruits. The corresponding gene was functionally expressed in yeast (*Saccharomyces cerevisiae*) and characterized for substrate specificity with a series of putative aromatic precursors with an aromatic vanilloid structure. Methylenedioxy bridge formation was only detected when feruperic acid (5-(4-hydroxy-3-methoxyphenyl)-2,4-pentadienoic acid) was used as a substrate, and the corresponding product was identified as piperic acid. Two alternative precursors, ferulic acid and feruperine, were not accepted. Our data provide experimental evidence that formation of the piperine methylenedioxy bridge takes place in young black pepper fruits after a currently hypothetical chain elongation of ferulic acid and before the formation of the amide bond. The partially characterized enzyme was classified as CYP719A37 and is discussed in terms of specificity, storage, and phylogenetic origin of CYP719 catalyzed reactions in magnoliids and eudicots.

## 1. Introduction

Plants are known for their remarkable structural diversity of specialized metabolites [1]. The resulting plethora of compounds is partly based on the evolution of cytochrome P450 enzymes (CYPs), heme-dependent monooxygenases that catalyze a wide variety of hydroxylation and monooxygenation reactions [2,3]. Therefore, CYPs play an essential role in the chemical modifications of all types of natural products, like terpenoids, phenylpropanoids, alkaloids, glucosinolates, or cyanogenic glycosides to name the most prominent ones [4]. Eukaryotic CYPs show an N-terminal membrane anchor and are known as ER-associated proteins of many biochemical pathways. They obtain electrons from NADPH-cytochrome P450 reductases (CPRs) usually required for catalytic activity.

Besides common reactions like oxidation and hydroxylation of a carbon skeleton, CYPs are also known for unusual reactions including oxidative C-C bond cleavage, methylenedioxy bridge formation, and phenol coupling [5]. Methylenedioxy bridges and phenol-couplings can be performed by the CYP719 clade within the heterogeneous CYP71 clan, classified as CYP719A [6,7], whereas phenol-coupling is performed by the CYP719B subclade, respectively [8]. The CYP719 clade is absent from Arabidopsis and rice but common in angiosperms comprising Ranunculales, Aristolochiales, and Proteales [3,9,10]. Most CYP719A members catalyze reactions in benzylisoquinolid alkaloid formation and are substrate and position specific. CYP719A1 from Japanese goldthreat (*Coptis japonica*) converts tetrahydrocolumbamine to tetrahydroberberine, the precursor of berberine [11]. Orthologues, CYP719A2 and CYP719A3 from *C. japonica* and greater celandine (*Chelidonium majus*) exhibit cheilantifoline synthase activity from scoulerine to cheilanthifoline and stylopine synthase activity from cheilanthifoline to stylopine [11,12] while CYP719A5 from California poppy (*Eschscholzia californica*) and CYP719A13 from Mexican prickly poppy (*Argemone Mexicana*), introduce two consecutive methylenedioxy bridges and convert scoulerine directly to cheilanthifoline [7,13]. In opium poppy (*Papaver somniferum)*, at least three CYP719A enzymes catalyze the formation of position specific methylenedioxy bridges in sanguinarine and noscapine biosynthesis [14,15]. Besides their presence in alkaloids, methylenedioxy groups are also known from phenylpropanoids, specifically lignans. CYP719A23 and CYP719A24 introduce methylendioxy bridges in podophyllotoxins from mandrake (*Podophyllum peltatum*) and May apple, *Podophyllum hexandrum* [16]. Stereoselective CYP81Q1 and CYP81Q3 from sesame (*Sesamum indicum*), also within the CYP71 clan catalyze similar reactions and convert pinoresinol to piperitol and sesamin in two consecutive steps in the biosynthesis of lignans [17]. Both enzymes share less than 25% amino acid sequence identity with the CYP719 clade [18]. Most recently, an unusual CYP719A26 was identified that catalyzes methylenedioxy bridge formation of kavalactones, potentially anxiolytic polyketide-derived styrylpyrones from kava (*Piper methysticum*) rhizomes [19].

*Piper nigrum*, black pepper, is known as one of the world’s most relevant spices due to its high content of piperine, a pungent alkaloid and phenolamide. Besides its daily use as a spice, numerous applications have been reported for piperine and related piperamides, as traditional medicine, but also with current pharmacological relevance [20,21]. Some of these effects are based on the interaction of piperine and the structurally related capsaicin from hot chili peppers (*Capsicum spec*.) with the human TRPV1 receptor, also termed vanilloid receptor, resulting in the observed burning sensation, triggering heat and nerve pain responses [22,23]. Although high concentrations of piperine and related piperamides are present in black pepper and related Piperaceae, only little is known about their biosynthesis (Figure 1). The two parts of the molecule are derived from independent biosynthetic pathways. The piperidine heterocycle appears essentially derived from L-lysine, as shown in labelling experiments performed with different stonecrops, *Sedum sarmentosum* and *Sedum acre* dating back 50 years ago [24,25]. The aromatic part of piperine is likely derived from the phenylpropanoid metabolism. In 1990, the activity of a piperoyl-CoA:piperidine-N-piperoyl transferase (piperine synthase) enzyme from crude black pepper shoot extracts was described which resulted in the formation of piperine by condensation of piperoyl-coenzyme A (piperoyl-CoA) with piperidine [26]. Most recently, two different piperoyl-CoA ligases, catalyzing the relevant CoA ester formation from piperic acid were identified and functionally characterized from immature fruits and leaves of black pepper [27,28].

The characteristic methylenedioxy bridge of the phenylpropanoid part of the molecule is one of the unsolved biosynthetic hallmarks in piperine formation. In principle, it can be introduced at the level of ferulic or feruperic acid, but also late as a final modification after amide bond formation at the level of feruperine Based on a transcriptional approach, using RNA sequencing data from different black pepper organs, combined with functional expression in yeast, we describe the identification and partial characterization of a fruit specific cytochrome P450 enzyme that catalyzes the formation of the 3,4-methylenedioxy group in the aromatic part of the piperine molecule.

## 2. Results

### 2.1. Identification and Cloning of PnCYP719 and PnCPR

Piperine is the dominant specialized metabolite already in maturing black pepper fruits, where it rapidly rises starting from 1 nmol/mg fresh weight around 20 days post anthesis to 10 nmol/mg around 40 days post anthesis (Figure 2). In mature fruits the piperine concentration further increases up to 100 nmol/mg fresh weight, 100 days post anthesis (Appendix A). No piperine accumulates in leaves and flowering spadices. The piperine content in mature roots, although less than in fruits, is noteworthy. Due to the abundance of piperine and the high transcript levels of the specific piperoyl-CoA ligase in black pepper fruits [27] we assumed that related genes of piperine biosynthesis are co-expressed or at least highly expressed at similar stages of fruit development. In addition, based on literature data, the 3,4-methylenedioxy bridge of piperine might be synthesized by an enzyme, either from the CYP719A or the CYP81Q clade. Therefore, de novo fruit transcriptome data sets that were generated at the IPB were screened for putative CYPs induced early during fruit ripening. Only a single CYP-sequence was identified in the fruit transcriptome among the top 30 most abundant sequences of fruits 20-30 days and 40-60 days post anthesis together with a series of phenylpropanoid related transcripts.

This single sequence showed highest similarity to cheilanthifoline and S-canadine synthases, both members of the CYP719A-clade [13]. Interestingly, no other P450-like transcript was identified in this fruit transcriptome. The transcript was also expressed 10-fold higher (transcripts per million) in immature fruits, compared to transcriptomes of leaves or flowering spadices. A single, full length transcript of 1500 bp could be assembled encoding a 56 kDa protein consistent with the expected sequence length and molecular masses of plant CYP719A-type enzymes. Fruit transcriptome data were additionally monitored for NADPH-dependent cytochrome P450 reductase (CPR) candidates. Again, a single full-length transcript with high similarities to Arabidopsis reductases, ATR1 and ATR2 (encoded by At4g24520 and At4g30210; http://www.tair.org) was identified. Total RNA from fruits, two months after pollination were extracted, cDNA synthesized, and the corresponding PnCYP719 and PnCPR encoding candidates were cloned using the modular Golden-Gate cloning system [29].

### 2.2. PnCYP719 Is Differentially Expressed in Black Pepper Organs

To confirm the assumption of fruit-specific expression of PnCYP719 and corroborate data from the RNA sequencing we analyzed the expression of this gene by RT-qPCR from the black pepper organs including flowering spadices, fruits (one and two months after anthesis), young leaves, and roots. The black pepper elF2B gene, encoding a fairly equally expressed elongation factor [27] was used as an internal reference gene. Results are shown in Figure 3. The gene encoding PnCYP719 was expressed preferentially during fruit development compared to leaves, flowers, or roots. Consistent with the presence of piperine in roots, transcript levels can also be detected in roots, compared to virtual absence of leaves and flowers. In the case of the corresponding reductase gene encoding PnCPR, a similar, yet less profound trend was observed for all organs. These data support the assumption that genes encoding putative piperine biosynthetic enzymes, like piperoyl-CoA ligase and this PnCYP719-like gene are preferentially and co-expressed during fruit development.

### 2.3. Heterologously Expressed PnCYP719 Is Substrate Specific

Functional definition of enzymatic activities requires annotation of substrate specificity. Activity of ER-localized plant derived CYP-sequences has successfully been obtained by expression in yeast [30,31,32]. The modular Golden Gate cloning system was used to simultaneously express two different reductases from Arabidopsis and black pepper, ATR1 and PnCPR, respectively in combination with PnCYP719. Product formation was monitored after isolating of microsomes from yeast cells. Alternatively, substrates were fed directly to cultured intact yeast cells.

Substrates were selected solely based on the presence of the aromatic vanilloid moiety and can be classified as C6-C5 (feruperic acid and feruperine), C6-C3 (coniferyl alcohol, coniferyl aldehyde, and ferulic acid), or C6-C1 (vanillin and vanillic acid) structures according to the length of the aliphatic carbon chain. Substrates with an extended aliphatic carbon chain, feruperic acid and feruperine, were considered as the most likely precursors, but are not commercially available. The amide feruperine was synthesized from feruperic acid, which chemical synthesis is recently described [27]. The structure was confirmed by ^1^H, ^13^C NMR, and mass spectrometry.

Detection of reaction products was based on the UV-absorbance and an expected mass deficit of the methylenedioxy group, caused by the loss of two hydrogen atoms of the product, versus the vanilloid-like substrate. Among all substrates tested, only feruperic acid was converted into piperic acid, harboring the required methylendioxy group. This reaction was performed with high efficiency by microsomes prepared from yeast in a protein-dependent manner (Figure 4). A steady increase can be observed from 50 µg to 250 µg total protein amount. The resulting product peaks showed the same retention time, expected mass signal of m/z 219.1 [M+H]^+^, and UV-spectrum with a maximum at 344 nm as a commercially available piperic acid standard. Identical results were obtained with both reductases, PnCPR and ATR1 (Figure 4). No signal was detected with microsomes isolated from wild-type and vector control transformed yeast. Feeding feruperic acid directly to transgenic whole yeast cell assay systems (fermentation), again resulted in piperic acid detectable in the supernatant, but with lower intensity compared to isolated microsomes. In all other cases, neither piperic acid nor the formation of any methylenedioxy group was observed (Table 1). Most importantly, we never detected a product signal (piperine) in the case of a second potential substrate, the amide feruperine even upon prolonged incubation times up to 24 h, although this compound was isolated from black pepper and other Piper species previously (Appendix A, [33]. No product formation (3,4-methylenedioxycinnamic acid) was also detected with ferulic acid (Appendix A).

When other vanilloid C6-C3 or C6-C1 structures were fed to yeast, no corresponding signals of a methylendioxy bridge formation, was observed. Instead several detoxification products were detected in yeast after prolonged incubation of 24 h. E.g. coniferyl alcohol was converted into vanillic acid m/z 171.1 [M+H]^+^ and also to ferulic acid m/z 195.1 [M+H]^+^. Coniferyl aldehyde was converted into ferulic acid. Vanillyl alcohol and vanillyl aldehyde were oxidized into vanillic acid. Similar observations were reported previously [34]. In summary, these data corroborate that PnCYP719 has exclusive substrate specificity for feruperic acid. Our data are consistent with the assumption that piperine in black pepper fruits is synthesized after chain elongation of ferulic acid in a reaction cascade starting from a C6-C5 precursor, feruperic acid to piperic acid, activation by piperoyl-CoA ligase, and subsequent piperine formation catalyzed by a piperine synthase [26,27].

### 2.4. Phylogeny and Structural Aspects of PnCYP719 and PnCPR

The PnCYP719 sequence was the only full-length cDNA in transcriptomes of all organs analysed with similarities to genes that encode possible methylenedioxy bridge-forming enzymes. The enzyme displays 95% identity at the amino acid level to the recently identified CYP719A26 from the rhizome of *Piper methysticum*, introducing a methylenedioxy bridge in kavalactones [19]. Yet, both enzymes apparently display different substrate and organ specificities.

Therefore, PnCYP719, although likely an orthologue of CYP719A26, was classified as CYP719A37 (http://drnelson.uthsc.edu/CytochromeP450.html) [35]. Its similarity to CYP179A26 from kava is illustrated in a resulting phylogenetic tree, which shows several representatives of the CYP719, CYP81, and CYP77 clades, all members of the diverse CYP71 clan (Figure 5). Both Piper sequences share roughly 50% identity to the branch of several classified CYP719A and CYP719B-like enzymes characterized from various benzylisoquinoline alkaloid producing Ranunculaceae and Papaveraceae, including tetrahydroberberine synthase from C. japonica, cheilanthifoline, canadine, and stylopine synthases from opium poppy (*P. somniferum*) and prickly poppy (*A. mexicana*), but also to salutaridine synthase CYP719B1 from *A. mexicana*. Up to now CYP719B1 is the only characterized member of the C-C-phenol-coupling enzyme in this clade [8], a reaction catalyzed also by members of the CYP80 clade [36]. The CYP719A26 and CYP719A37 from both Piper species also cluster to CYP719-like sequences from early diverging angiosperms. Up to 55% sequence identity is observed to a group of CYP719-like sequences from stout camphor (*Cinnamomum kanehirae*), a member of the Lauraceae. Only two out of six sequences are listed. A single CYP719-like sequence from sacred lotus, Nelumbo nucifera is also among the closest putative orthologues. The sequence with the highest identity outside the CYP719 clade encodes CYP77A4 from Arabidopsis, that catalyzes epoxidation of fatty acids [31]. Members of the more promiscuous CYP81 clade are distinct from the CYP719 clade, although in principle the same type of reaction can be performed. CYP719A37 shows typical eukaryotic structural features like the K-helix, the aromatic- as well as the heme-binding domain at the C-terminus and a hydrophobic N-terminal membrane anchor, corroborating presumed ER-association. It also contains the characteristic G300L and T304S substitutions distinguishing the CYP719A clade from the CYP77 clade (Appendix A).

The black pepper reductase, PnCPR identified in parallel from the fruit transcriptome, showed the expected amino acid sequence identities to oxidoreductases, 63% to ATR1 and 68% to ATR2, respectively from Arabidopsis. PnCPR shows all conserved motifs of a classical CPR including FMN-PPi-, FMN-isoallozazine-, substrate-, FAD-PPi-, and NADPH-ribose binding domains.

## 3. Discussion

Cytochrome P450s provide a plethora of enzymatic reactions and are essentially responsible for diversification of specialized pathways throughout the plant kingdom [3]. The pathways to piperamide formation are still puzzling although piperine is known for 200 years and initial biosynthetic experiments date back several decades ago. A combination of bioinformatic, synthetic, and molecular tools enabled us to add a coveted piece to this puzzle. Our data experimentally verify previous assumptions that the methylenedioxy bridge of piperine is introduced at the level of feruperic acid, after C_2_-extension of the aliphatic carbon chain. This plausible cascade of biosynthetic reactions, starting by oxidation of the vanilloid structure of feruperic acid, subsequent activation of piperic acid by piperoyl-CoA ligase [27], terminates a phenylpropanoid derived cascade reactions resulting in the formation of an alkaloid after presumed amide formation by piperine synthase [26]. The identification of CYP719A37 was guided by the high abundance of a single CYP-transcript in maturing green fruits and its obvious similarity to enzymes catalyzing methylenedioxy bridge formation in benzylisoquinoline biosynthesis in the Ranunculaceae and Papaveraceae [7,13,37]. In previous cases, CYP-based co-expression analysis resulted in the successful elucidation of complete pathways of specialized metabolism, like sporopollenin biosynthesis [38]. CYP719A37 and piperoyl-CoA ligase [27] appear co-expressed with a series of other putative pathway genes which are currently analyzed in our group. These include candidate genes for the enigmatic C2-elongation of ferulic acid as well as acyltransferases that might encode piperine synthase. Combined with the recent annotation of the black pepper genome [39] and our differential transcriptome data set, characterization of the remaining steps of piperine biosynthesis is anticipated in due course [40].

Yeast microsomes provide a reliable source and activity for heterologous expression of CYP719-like enzymes [7,41]. The CYP719 clade contains several alkaloid biosynthesis enzymes of high substrate and even regiospecificity [6,7,13,14]. Consistent with previous assumptions, CYP719A37 also appears unique in its substrate preference for a minimum aliphatic chain length of five carbons. It seems likely that substrates, resulting in piperamides with slightly longer chain lengths like piperettine (C_6_-C_7_), a minor compound in black pepper fruits, could also be oxidized by the enzyme. Piperamides, with extended C_13_-aliphatic carbon chains like guineensine detected in *Piper guineense* seeds [42] may require CYP719A37 orthologues with different substrate preferences. Functional expression of these orthologues from various Piperaceae will reveal critical amino acid required for carbon chain length specificity. In opposite to nicotine or pyrrolizidine alkaloids, which are usually synthesized in the roots [43,44], we currently assume that the biosynthesis of piperine takes place predominantly in the fruits. This is based on the high levels on piperine and the corresponding biosynthetic enzymes, also including the decisive piperine synthase [40]. Nevertheless, the presence of piperine and a low expression level of the gene encoding CYP719A37 also suggest a role for the compound in roots, presumably for defence.

The extremely high piperine content in fruits requires specific channeling, compartmentation, and specialized cells for storage and deposition. Several potential storage compartments of different size and origin from differentiated plastids, to vacuolar or ER-derived micro-compartments, up to large glandular compartments have been the focus of recent investigations to explain storage of insoluble or unstable compounds up to molar concentrations [45]. Natural deep eutectic solvents (NADES), consisting of defined mixtures of different sugars, organic acids, choline, or betaine not only increase stabilization and solubilization of labile and otherwise insoluble metabolites, but also conserve activity and organization of the corresponding biosynthetic enzymes, as shown recently for the biosynthesis of dhurrin and cannabinoids. [46,47,48]. Investigations of these or other factors contributing to channeling of pathway intermediates like piperic acid and subsequent storage of piperine in fruits will be a rewarding area for future investigations.

CYP719 enzymes are members of the large CYP71 clan, which is considered an innovation of angiosperms [3]. Yet, methylenedioxy bridges and *CYP719* genes appear to be missing in early diverging angiosperms, Amborellales, Nymphaeales, and Austrobaileyales, (ANA-clade). The genomes of *Amborella trichopoda* and the recently sequenced water lily (*Nymphaea cordata*), both considered sister lineages to all extant angiosperms do neither harbor any CYP719-like sequences nor contain any benzylisoquinoline alkaloids or methylenedioxy bridge containing lignans [49,50,51,52]. This indicates that the CYP719 clade developed after the diversification of the ANA-clade from magnoliids and eudicots. Consistent with our transcriptome data, recently published black pepper genome data, and the genome annotation of sacred lotus, only a single CYP719 sequence is present in both magnoliid genomes [39,53], whereas stout camphor (Lauraceae) contains several CYP719 genes [54]. CYP719-gene duplication events occurred in the Ranunculaceae and Papaveraceae [6,7,14,15,55]. The CYP719A26 and CYP719A37 sequences of kava and black pepper are likely derived from a common ancestor. Kava is considered a group of sterile, fruitless cultivars of *Piper wichmannii*, endemic to the Polynesian islands and it is geographically separated from black pepper [56]. Its unique blend of styrylpyrones is exclusively synthesized in the roots [19]. It is well known, that nearly identical P450 sequences may catalyze different reactions. A single change from F209L turned coumarin 7-hydroxylase into testosterone 15α-hydroxylase [57] and an F363I substitution converted limonene-C6-hydroxylase of spearmint to peppermint limonene-C3-hydroxylase which otherwise are 70% identical [58]. Therefore, during diversification of the Piperaceae and geographical isolation, *P. nigrum* and *P. methysticum* sequences may have developed different substrate preferences independently from each other. While the kavalactone substrates are not available, cloning and functional expression of the kava enzyme will be performed to be tested with piperine precursors in future studies. The methylenedioxy group is widely spread among the pantropic Piperaceae and Piperales. It appears characteristic for the plethora of piperamides, but is usually not detected in accompanying phenylpropenes and lignans of the same species [59]. Rapidly increasing sequence information combined with transcriptome and genome mining for CYP719-orthologues in the Piperales with more than 4000 species [60] should delineate the structure-function relationship of the *Piper* CYP719 subclade, shed light on functional aspects of individual compounds in vivo, and contribute to the development of new biocatalysts for biotechnological and pharmacological applications [61,62].

## 4. Materials and Methods

### 4.1. Plant Material, RNA Extraction and cDNA Synthesis

Cuttings from black pepper plants were obtained from the Botanical Garden of the University of Vienna (Austria) from plants collected in 1992, IPN No. LK-0-WU-0014181 and grown under temperature and humidity-controlled glasshouse conditions as described previously [27]. Transcriptome data were obtained from fruits, leaves, and flowering spadices, and have been submitted to ArrayExpress (http://www.ebi.ac.uk/arrayexpress/experiments/E-MTAB-9029). For RT-qPCR analysis, RNA from fruits at different developmental stages, from flowering spadices, young leaves, and roots were harvested and directly ground in liquid nitrogen via a ball mill (Retsch, Germany). RNA was extracted by Nucleospin^®^ RNA Plus Kit (Machery-Nagel, Düren, Germany) according to manufacturer instructions and subsequently reverse-transcribed by the Maxima H Minus First Strand cDNA Synthese Kit (Thermo Fisher Scientific, Dreieich, Germany).

### 4.2. Chemical Synthesis of Feruperic Acid and Feruperine

All chemicals, including substrates with a vanilloid structure were purchased from Merck (Sigma, Darmstadt, Germany). The chemical synthesis of feruperic acid (Figure 1) has been described in detail recently [27]. The alternative substrate feruperine (Figure 1) was synthesized according to [23]. Briefly, 100 mg feruperic acid (100 mg) were mixed with DIC (N,N′-diisopropylcarbodiimide) (76 µL) and HOBt (hydroxybenzotriazole monohydrate) (73.4 mg) in dry dichloromethane (25 mL) for 10 min at room temperature followed by the addition of piperidine (46 µL). The resulting mixture was stirred until the starting material was completely consumed (TLC: dichloromethane 3:2 ethyl acetate). The crude product of the reaction was purified on silica gel (dichloromethane/ethyl acetate 3:2). Final product: 75 mg, yield: 57%. Feruperine: ^1^H NMR (CDCl_3_, 400 MHz) δ (ppm): 1.58 (m, H-3” and H-5”), 1.66 (m, H-4”), 3.61–3.56 (m, H-2” and H-6”), 3.91 (s, H-7′), 6.43 (d, *J* = 15.2 Hz, H-2), 6.76 (overlapped signal, H-4 and H-5), 6.90 (d, *J* = 8.0 Hz, H-5′), 6.94 (d, *J* = 1.6 Hz, H-2′), 6.97 (dd, *J* = 8.0, 1.6 Hz, H-6′), 7.42 (ddd, *J* = 15.2; 5.5 Hz, H-3). ^13^C NMR (CDCl_3_, 100 MHz) δ (ppm): 24.6 (C-4”), 26.5 (C-3”), 26.6 (C-5”), 43.2 (C-2”), 46.8 (C-6”), 55.8 (C-7′), 108.8 (C-2′), 114.8 (C-5′), 119.4 (C-2), 121.0 (C-6′), 124.7 (C-4), 128.9 (C-1′), 138.7 (C-5), 142.8 (C-3), 146.6 (C-3′), 146.8 (C-4′), 165.5 (C-1). MS: *m/z*: 286.4 [M-H] for C_17_H_21_NO_3_.

### 4.3. Preparation of Black Pepper Methanolic Extracts

Methanolic extracts were generated from greenhouse grown black pepper fruits, spadices, leaves, and roots. Tissues were frozen, macerated in a ball mill (Retsch), taken up at a concentration of 100 mg/mL in 90% methanol, ultrasonicated for 5 min, and centrifuged for 2 min at 20,000 g. Roots and fruits were diluted to 10 mg/mL. All samples were analysed by the Waters Alliance HPLC equipped with a UV-detector and a QDA mass detector on a Nucleosil RP_8_, 5 µm, 12 cm (Machery-Nagel) equipped with the Empower III software package. A gradient of water with 0.1% (*v*/*v*) formic acid (solvent A) and acetonitrile (solvent B) was used. All extracts were separated by a gradient from 30% to 90% solvent B in solvent A within 10 min with a flow rate of 0.8 mL min^-1^ and compared to a standard of 100 pmol/µL of piperine. 5 µL or 10 µL of standard and extracts were injected.

### 4.4. Cloning of PnCYP719A and PnCPR Genes

A black pepper transcriptome data set was screened for CYP and CPR-orthologous sequences specifically and highly expressed in black pepper fruits. Full length candidate genes encoding *Pn*CYP719A37 and *Pn*CPR were selected, amplified with gene specific primers (Eurofins Genomics, Germany; Table 1) and cloned according to established Golden-Gate-cloning protocols for protein expression in yeast [29,63,64]. Briefly, fragments encoding *Pn*CYP719 and *Pn*CPR were amplified and internal BsaI/BpiI restriction sites removed simultaneously by a silent point mutation at assembly *level -1*. At *level 0* the fragments were assembled into an entry vector. Plasmids were purified by Nucleospin^®^ plasmid purification kit (Machery-Nagel) and amplicons sequenced by Eurofins Genomics. Subsequently, the amplicons encoding CYP719A37 were combined with the synthetic galactose-inducible promoter 6 (*Gal6*) and the tPGK-1-terminator (assembly *level 1*). At assembly *level M*, the *CYP719A37* amplicon finally was combined with one out of two reductases, either *PnCPR* or *ATR1* as described previously. *Gal6* and a fragment, encoding ATR1 for expression in yeast were kindly provided by Alain Tissier and Ulschan Bathe (Leibniz-Institute for Plant Biochemistry, IPB, Germany) [32]. Cloning vectors contained an origin of replication for *E. coli* (*ColE1*) and *S. cerevisae* (2 µm *ori*). Kanamycin (level -1), ampicillin (level 1), and spectinomycin (level 0 and level M) were used as *E. coli* selection markers. The *URA3* gene, encoding orotidine 5-phosphate decarboxylase was used as a marker for selection of positive yeast transformants. Genbank accession numbers of CYP719A37 and *Pn*CPR are MT643912 and MT643913, respectively.

### 4.5. Transient Gene Expression in S. Cerevisiae INVSc1

The expression plasmids, *PnCPR:CYP719A37* and *ATR1:CYP719A37* were subsequently transformed into yeast strain INVSc1 (Thermo Fisher Scientific) according to standard protocols. A positive colony was inoculated in 5 mL Ura3-medium with 2% glucose and grown at 30 °C and 170 rpm overnight. This pre-culture was used to inoculate 200 mL Ura3-medium supplemented with 2% glucose and grown at 30 °C and 170 rpm for 24 h. Cells were recovered by centrifugation (4000× *g*, 10 min), suspended in yeast extract-peptone medium with 2% galactose, and grown over night under identical conditions either for transient yeast cell assays or for preparation of microsomal fractions.

### 4.6. Characterization of Enzyme Activities in Transient Yeast Cell Assays

Following expression, yeast cells were centrifuged and adjusted to a final OD_600_ of 50 in 50 mM HEPES buffer (pH 7.5). For activity measurements 95 µL re-suspended yeast cells were incubated with 5 µL of 5 mM putative substrate at 30 °C, on a shaker at 1,000 rpm overnight. The cells were centrifuged (25,000× *g* for 20 min). The supernatant was separated and, in a 1:1 ratio, mixed with 100 µL of a mixture of four parts (*v*/*v*) methanol and one part (*v*/*v*) 20% formic acid. The cells were incubated again at 30 °C, 1000 rpm, and overnight for product extraction. The initial supernatants and the methanolic cell extracts were analyzed for product formation by LC-MS.

### 4.7. In Vitro CYP Assay with Microsomal Fractions

Microsomal fractions were prepared according to [32]. Briefly, yeast cells were crushed with glass beads in buffer containing sorbitol and BSA. Microsomes were obtained by centrifugation at 100,000 for 2 h and taken up in 50 mM Tris/HCl 7.5, 1 mM EDTA, 30% glycerol. Protein concentration of the microsomal fraction was determined by Bradford protein assay (Bio-Rad, Feldkirchen, Germany) and stored at −80 °C. Assays were run in triplicates. A single assay contained microsomal preparations of 50 µg–250 µg crude protein, 250 µM of putative substrate, 1 mM NADPH and 50 mM HEPES buffer (pH 7.5) in a total volume of 600 µL and was incubated for 2 h (in the case of feruperic acid, feruperine and ferulic acid) and up to 24 h in the case of other substrates at 30 °C and 90 rpm. The reaction was stopped by adding 20 µL of 20% formic acid and incubated on ice for 10 min. The protein was precipitated by centrifugation at 25,000× *g* for 15 min. The reactions were extracted with 600 µL ethyl acetate. The organic phase was concentrated into 100 µL 100% methanol and product formation monitored simultaneously by UV- and QDA-mass detection (Waters). 10 µL of the reactions were analysed on a Nucleoshell^®^ RP18, 2.7 µm, 5 cm (Machery-Nagel) with water 0.1% (*v*/*v*) formic acid (solvent A) and acetonitrile (solvent B). All reactions and standards were separated by a gradient from 10% to 90% solvent B in solvent A within 7 min and a flow rate of 0.6 mL min^−1^. Putative substrates and products were identified by retention time, UV-absorption, and ESI-MS in positive ionization mode and compared to synthesized or commercially available standards.

### 4.8. RT-qPCR of PnCYPA37 and PnCPR Transcripts

Expression pattern of the transcripts encoding CYP719A37 and PnCPR were analyzed by RT-qPCR based on cDNA from fruits (one month and two months after pollination), flowering spadices, young leaves, and roots. 200 ng of RNA were used for cDNA synthesis. The RT-qPCR was performed with a qPCR Mix EvaGreen^®^ No Rox (Bio&Sell, Feucht, Germany) with 3 µL of 1:10 diluted cDNA and 2 pmol of each primer (Table 1) in a 10 µL reaction. The reaction was recorded by CFX ConnectTM Real-Time System (Bio-Rad, Munich, Germany). The black pepper *elF2B* gene was used as a reference [27]. The experiment was performed with three technical and three independent biological samples per tissue.

### 4.9. Data Availability and Deposition

NCBI accession numbers and gene identifiers of Figure 5 are listed in Appendix A. RNA-Seq data were stored in array express and will be accessible under the following link: http://www.ebi.ac.uk/arrayexpress/experiments/E-MTAB-9029.

## 5. Conclusions

In summary, we identified an enzyme catalyzing the critical methylenedioxy bridge formation in piperine biosynthesis prior to activation by a CoA-ligase. This step experimentally verifies a hypothetical pathway based on the discovery of a piperine synthase activity that links the phenylpropanoid metabolism to amino acid metabolism, resulting in a pungent alkaloid and the molecular structure of the world’s most prominent spice since antiquity.

## Figures and Tables

**Figure 1 plants-10-00128-f001:**
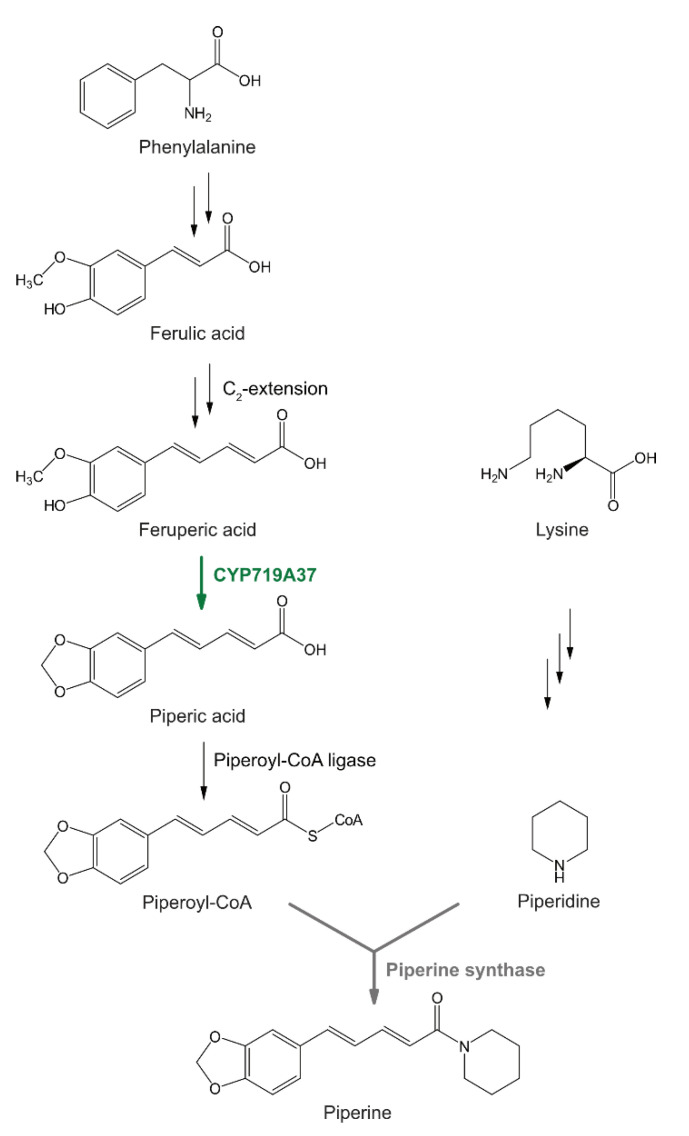
Presumed biosynthesis of piperine, derived from an aromatic part, piperoyl-CoA, and lysine biosynthesis, contributing the piperidine heterocycle. A CYP719-like enzyme described in this report, *Pn*CYP719A37, catalyzes the formation of a methylenedioxy bridge in the vanilloid signature of feruperic acid to piperic acid.

**Figure 2 plants-10-00128-f002:**
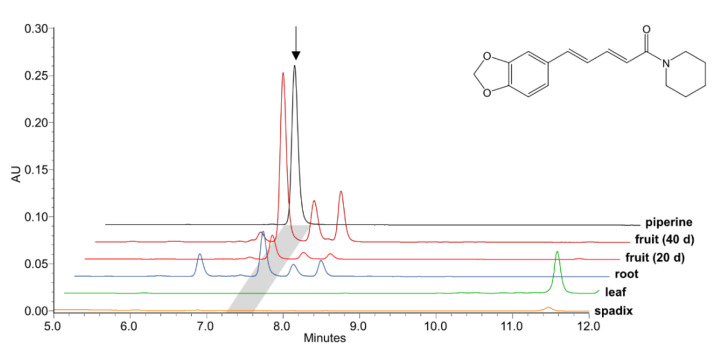
Piperine levels in young black pepper fruits 20 and 40 days after anthesis as compared to flowering spadices, leaves, and roots. Piperine and related amides are virtually absent in flowering spadices and leaves, whereas they also occur at a lower level in roots. Signals were compared to 500 pmol of a piperine standard. Levels were monitored by LC-MS at 340 nm and at m/z 286.1 [M+H]^+^. The piperine peak is marked in grey and its structure is shown.

**Figure 3 plants-10-00128-f003:**
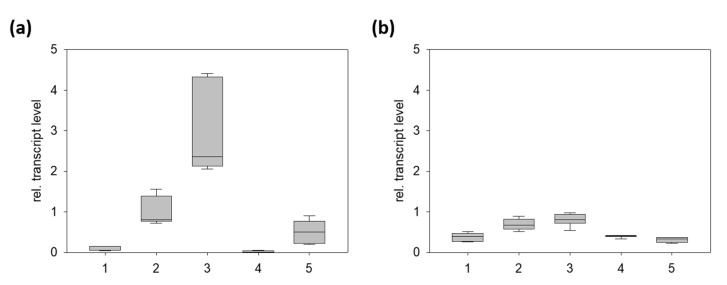
Relative transcript levels of the gene encoding CYP719A37 (**a**) and PnCPR (**b**) in different organs of P. nigrum using elF2B (TR23978) as the reference gene [27]. Each box represents nine data points, consisting of three biological replicates with three technical replicates. 1, flowering spadices; 2, fruits (20–30 d); 3, fruits (40–60 d); 4, young leaves; 5, roots.

**Figure 4 plants-10-00128-f004:**
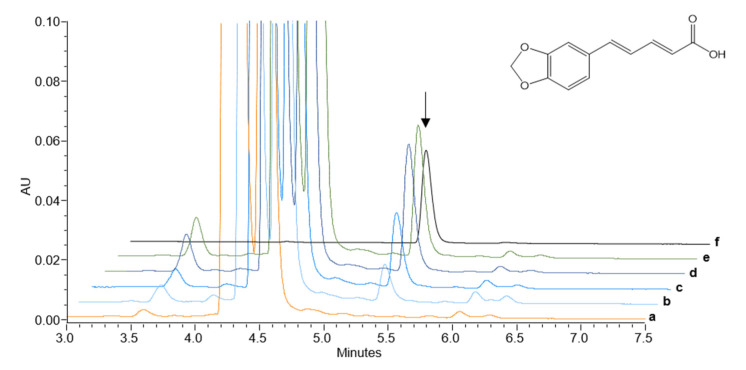
HPLC-UV-MS analysis of microsomal fractions of PnCYP719 and PnCPR transformed yeast incubated with feruperic acid for 2 h. Chromatograms showing piperic acid production by three different amounts of microsomal extracts (**b**, 50 µg; **c**, 100 µg; **d**, 250 µg total protein with PnCPR) and reductases (**e**, 250 µg with ATR1). Chromatograms of empty vector controls (**a**) and of 10 pmol of piperic acid (**f**) as reference product are also shown. UV signals at 344 nm and MS signals of the corresponding single mass ion for piperic acid (m/z 219.1, [M+H^+^]^+^) were recorded. The product piperic acid is marked with an arrow and its structure illustrated on the right.

**Figure 5 plants-10-00128-f005:**
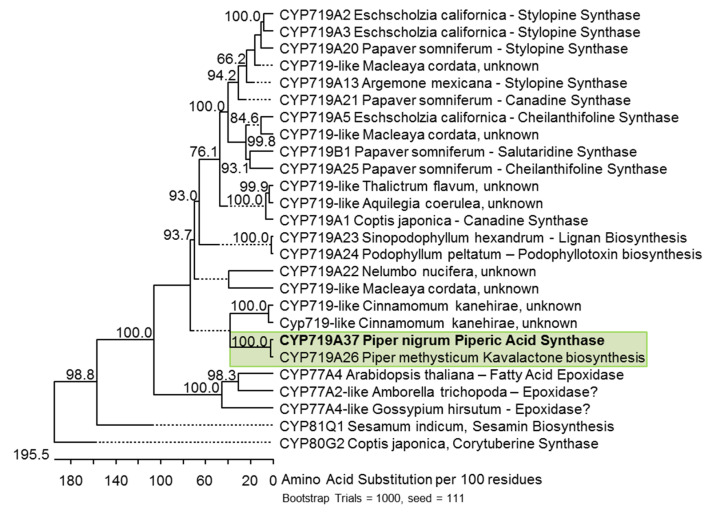
Bootstrapped unrooted cladogram of the black pepper CYP719A37 protein sequence among 25 selected CYP-sequences which show methylenedioxy bridge formation (CYP719A, CYP81Q), phenol coupling (CYP719B, CYP80G2), or epoxide formation (CYP77). Subclade of Piperaceae enzymes is boxed in green. The cladogram was created using the Clustal V algorithm. Accession numbers are listed in Appendix B.

**Table 1 plants-10-00128-t001:** List of aromatic substrates with a vanilloid signature tested in the microsomal PnCYP719 assays up to 24 h and corresponding masses [m/z] of substrates and products.

Substrate [*m/z*]	Product [*m/z*]	Activity [pkat/mg Crude Protein]
**Feruperic acid [219.1]**	**Piperic acid [217.1]**	**0.92+/−0.05**
Feruperine [287.1]	Piperine [285.1]	n.d. ^2^
Ferulic acid [194.1]	3,4-methylenedioxycinnamic acid [192.1]	n.d.
Coniferylaldehyde [178.1]	3,4-methylenedioxycinnamylaldehyde ^1^ [176.1]	n.d
Coniferylalcohol [180.1]	3,4-methylenedioxycinnamylalcohol ^1^ [178.1]	n.d.
Vanillic acid [168.1]	Piperonylic acid [166.1]	n.d.
Isovanillic acid [168.1]	Piperonylic acid [166.1]	n.d.
Vanillylaldehyde [152.1]	Piperonal ^1^ [150.1]	n.d.
Vanillylalcohol [154.1]	Piperonylalcohol [152.1]	n.d.

^1^ No standard available, identification would be based on expected mass signals. ^2^ n.d. not detected.

## Data Availability

Data are stored permanently at www.radar-service.eu. The DOI number: 10.22000/381 was assigned.

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
