# Peer review of "Piper nigrum CYP719A37 Catalyzes the Decisive Methylenedioxy Bridge Formation in Piperine Biosynthesis"

_plants, 2021, doi:10.3390/plants10010128_

Round 1
Reviewer 1 Report
The manuscript identifies CYP719A37 as the P450 catalyzing methylenedioxy bridge formation in piperine formation. Black pepper is one of the worlds most important spices due to its high content of piperine. It is great that yet an additional step in synthesis of this coveted spice has now been resolved. Evolutionary aspects of the discovery are well described. The authors are using a very modest language to describe this major discovery.
The outline of the manuscript is clear, appropriate and state-of-the-art technologies are used, and data are presented with proper statistic analyses. The CYP719A37 showed high substrate specificity, an assessment made possible based on chemical synthesis of putative key substrates.
Minor edits:
line 44-45 language not clear
line 88: move "most recently" to the front of sentence
line 131 and other places: Pn to be written in italics
line 254: relevant should be substituted with the word coveted or key to emphasize the importance of the discovery
line 261: substitute facilitated with guided
line 281. The piperine content of the seeds is around 3% of the fresh weight and thus even higher on a dry weight basis. The authors may want to discuss that structurally similar cannabinoids as well as of some of their biosynthetic enzymes are co-stored in in trichome glands together with a natural deep eutectic solvent (Plant Science 284:108 (2019); Nat Prod Rep 35: 1140 (2018); Phytochemistry 170: 112214 (2020)
I recommend the manuscript to be accepted
Reviewer 2 Report
The article by Schnabel et al., titled "Piper nigrum CYP719A37 catalyzes decisive methylenedioxy bridge formation in piperine biosynthesis", describes the results of the multimodal study of a particular step in a secondary metabolite, piperine, biosynthetic pathway.
The paper is written clearly and is easy to read. The study described in the paper is designed on a high scientific level. A combination of transcriptomics, molecular cloning and heterologous expression is applied to decipher the metabolomic enigma of the methylenedioxy bridge formation during piperine biosynthesis.
Honestly, apart from minor spellcheck, this article is ready for the publication as it is.
Minor suggestions:
Lines 33, 35, 44, 45, 47, 52, 73, 77 82, 88 and so on have round brackets that are not closed properly. It is probably a technical issue due to reference manager program bug.
